# Quality and Metabolomics Analysis of *Houttuynia cordata* Based on HS-SPME/GC-MS

**DOI:** 10.3390/molecules27123921

**Published:** 2022-06-18

**Authors:** Shuai Qi, Lingyan Zha, Yongzheng Peng, Wei Luo, Kelin Chen, Xin Li, Danfeng Huang, Dongmei Yin

**Affiliations:** 1College of Ecology and Engineering, Shanghai Institute of Technology, Shanghai 201418, China; don_shuai@163.com (S.Q.); weichanyi183@163.com (W.L.); chenkelin@126.com (K.C.); 15543358779@163.com (X.L.); 2College of Agriculture and Biology, Shanghai Jiao Tong University, Shanghai 200240, China; zhaly2013@163.com (L.Z.); pengyongzheng@sjtu.edu.cn (Y.P.)

**Keywords:** *Houttuynia cordata*, medicinal ingredients, HS-SPME-GC-MS, metabolomics analysis, eating quality

## Abstract

*Houttuynia cordata* is a medicinal and edible plant with a wide biological interest. Many parts were discarded due to various modes of consumption, resulting in resource waste. In this study, a comprehensive study was conducted on various edible indicators and medicinal components of *Houttuynia cordata* to understand its edible and medicinal value. The edible indexes of each root, stem, and leaf were determined, and the metabolites of different parts were investigated using the headspace solid-phase micro-extraction technique (HS-SPME-GC-MS). The differential metabolites were screened by orthogonal partial least squares discriminant analysis (OPLS-DA) and clustering analysis. The results of the study showed that the parts of *Houttuynia cordata* with high edibility values as a vegetable were mainly the roots and leaves, with the highest vitamin C content in the roots and the highest total flavonoids, soluble sugars, and total protein in the leaves. The nutrient content of all the stems of *Houttuynia cordata* was lower and significantly different from the roots and leaves (*p* < 0.05). In addition, 209 metabolites were isolated from *Houttuynia cordata*, 135 in the roots, 146 in the stems, 158 in the leaves, and 91 shared metabolites. The clustering analysis and OPLS-DA found that the parts of *Houttuynia cordata* can be mainly divided into above-ground parts (leaves and stems) and underground parts (roots). When comparing the differential metabolites between the above-ground parts and underground parts, it was found that the most important medicinal component of *Houttuynia cordata*, 2-undecanone, was mainly concentrated in the underground parts. The cluster analysis resulted in 28 metabolites with up-regulation and 17 metabolites with down-regulation in the underground parts. Most of the main components of the underground part have pharmacological effects such as anti-inflammatory, anti-bacterial and antiviral, which are more suitable for drug development. Furthermore, the above-ground part has more spice components and good antioxidant capacity, which is suitable for the extraction of edible flavors. Therefore, by comparing and analyzing the differences between the edible and medicinal uses of different parts of *Houttuynia cordata* as a medicinal and food plant, good insights can be obtained into food development, pharmaceutical applications, agricultural development, and the hygiene and cosmetic industries. This paper provides a scientific basis for quality control and clinical use.

## 1. Introduction

*Houttuynia cordata*, widely distributed in Asia and North America, is a perennial herbaceous plant with high medicinal value [1]. It is a homology of medicine and food [2]. Various components of *Houttuynia cordata*, especially quercetin and kaempferol, have the potential to treat COVID-19 infection and COVID-19-induced cytokine storm by targeting multiple proteins [3]. In addition, it also has anti-HSV-1, anti-influenza virus, and anti-HIV-1 properties, as well as other viruses [4,5,6]. The main medicinal active components of *Houttuynia cordata* are volatile oil and flavonoids [7]. Extracts have antibacterial properties [8] and anti-inflammatory [6], antiviral [9], antioxidant [10], anti-hypertension [11], anti-mutagenic [12] and antibacterial [8] pharmacological effects. *Houttuynia cordata* injection has anti-inflammatory and antipyretic effects and can treat cancer, cough, dysentery, enteritis, and fever [11,13].

Many factors influence the differences in the medicinal components of plants, including production area [14], parts [15], cultivation methods [16], and environmental factors [17]. Different parts of many Chinese herbs have different uses, and the relative contents of medicinal ingredients vary greatly; ginseng fibrous roots are rich in ingredients and high ginsenosides contents, and different parts of the buckwheat plant have different polyphenolic and sugar compounds [15,18], and different parts of the flower contain different volatile components [19]. As wild vegetables, they can be divided into two types. The first type is sprouts, which are mainly eaten as tender leaves and stems, and their roots can be eaten in a salad and stir-fry. However, different eating methods result in the waste of parts of *Houttuynia cordata*. The utilization value of different parts of *Houttuynia cordata* is different, and the juice of *Houttuynia cordata* leaves can treat some diseases, including cholera and dysentery [20]. The extracts of roots and leaves have antioxidant properties and hemolytic activity [16]. With the continuous development of the medicinal potential of *Houttuynia cordata*, wild resources have been unable to meet the needs of the market, and the value of different parts should be rationally utilized. Therefore, it is of great significance to study the difference between the edibility value and medicinal value of different parts of *Houttuynia cordata*, which can be used effectively in different parts of *Houttuynia cordata*.

At present, GC-MS is the basis of the analysis of plant volatiles [21]. The excessive operation of the traditional GC method tends to lose some compounds and degrade volatiles [22]. Headspace solid-phase micro-extraction (HS-SPME) is a new technology; its main advantage is that it can eliminate interference from the sample matrix [23]. The distribution of compounds in different parts of *Houttuynia cordata* could be observed more accurately and intuitively by HS-SPME. The differential metabolites in different parts of *Houttuynia cordata* were analyzed for the first time. The differential metabolites were screened by OPLS-DA analysis, the cluster analysis of metabolites in different parts was conducted, and the potential correlation between volatile components was analyzed.

The main effective component of *Houttuynia cordata* was 2-undecanone [24]. In this study, based on the screening of differential metabolism, we found that 2-undecanone was mainly distributed in the underground parts of *Houttuynia cordata*. The edible and medicinal values of the different parts vary greatly, with the highly potent underground parts being used for the manufacture of drugs and the low-edibility-value stems being used as fodder. It is important to determine the composition of different parts to provide guidance for the rational use of different parts of *Houttuynia cordata* so as to maximize the medicinal and edibility value of *Houttuynia cordata*.

## 2. Results and Analysis

### 2.1. Analysis of Nutritional Quality of Different Parts

The total protein and VC contents in the stems of *Houttuynia cordata* were much lower than those of the roots and leaves (Figure 1A,B), with significant differences (*p* < 0.05). The content of soluble sugars in different parts showed significant variability, specifically, leaves > stems > roots (Figure 1C). The highest content of total flavonoids was found in leaves, followed by roots and stems, with significant differences (Figure 1D). In terms of the edible indexes of *Houttuynia cordata*, the highest edibility value was found in the leaves, followed by the roots, and the stems were the worst.

### 2.2. Identification of Volatile Compounds in Houttuynia cordata

The relative peaks of each ion were analyzed with the National Institute of Standards and Technology (NIST) database for substance characterization and combined with the standard mass spectra to determine the chemical components. The total ion flow diagrams of the root, stem, and leaf are shown in Figure 2A–C. The volatiles were analyzed and processed, the relative contents of metabolites were calculated by the area normalization method, and a total of 209 compounds were identified in *Houttuynia cordata*. Among them, 135 volatiles were isolated from the roots, 146 volatiles from the stem, and 158 volatiles from the leaves of *Houttuynia cordata*. There were 24 aldehydes, 24 esters, 22 alkanes, 20 alcohols, 19 alkenes, 11 ketones, 7 acids, 4 benzenes, 1 amide, 1 furan, 1 nitrile, and 1 pyrazine. The volatiles in the root, stem, and leaf accounted for 64.59%, 69.86% and 75.60% of the total volatiles, respectively, and the leaves contained the most metabolite species.

In Figure 3A, a total of 91 identical metabolites were found in different parts of *Houttuynia cordata*, with 22 metabolites unique to the roots and leaves and only 2 metabolites unique to the stems. In Figure 3B, 135 volatiles were detected in the roots, among which 20 olefins, 19 esters, 15 alcohols, 14 aldehydes, 12 alkanes, 7 acids, 6 ketones, 3 benzenes, 1 amide, 1 nitrile, 1 pyrazine and 35 others were accounted for. Olefins accounted for the largest proportion of volatiles in the roots, followed by ketones and lipids; 146 volatiles were detected in the stems, including 19 esters, 18 alkanes, 17 aldehydes, 15 olefins, 15 alcohols, 10 ketones, 6 acids, 3 benzenes and 38 other species. The highest relative contents were olefins, aldehydes, alkanes, and esters; 158 volatile substances were detected in the leaves of *Houttuynia cordata*, including 22 alkanes, 21 aldehydes, 18 lipids, 18 alcohols, 9 olefins, 8 ketones, 7 acids, 3 benzenes, 1 nitrile, 1 furan and a total of 38 others. The relative contents of aldehydes, alcohols, and olefins were the highest in total leaf volatiles.

### 2.3. Multivariate Statistical Analysis of Metabolomics

#### 2.3.1. Principal Component Analysis (PCA)

Principal component analysis (PCA) presents the characteristics of metabolomic multidimensional data through several principal components so that the variability of different groups can be observed through PCA plots. PCA was performed on the metabolites of different parts of *Houttuynia cordata* (PC1: 46.2%, PC2: 36.5%). As can be seen in Figure 4, the metabolites of different parts were separated on the first and second principal components, and there was significant variability in the metabolites of different parts. In the figure, each sample of each group is clustered into groups, indicating high data reproducibility and stability. The grouping of different parts was obvious, with roots dominating both the first and second principal components, leaves mainly distributed on the negative half-axis of the first principal component, and stems mainly on the negative half-axis of the second principal component.

#### 2.3.2. Cluster Analysis

The distribution characteristics of the metabolites of different parts of *Houttuynia cordata* can be shown using a clustering heat map. The clustering heat map shows (Figure 5) that there is significant variability in the groups of different parts of *Houttuynia cordata*, which are divided into two major clusters in total, one for above-ground parts (roots) and one for underground parts (stems and leaves). The replicate groups of different parts clustered into one cluster, indicating good biological replication and high confidence in the data. Therefore, this study focused on the differential metabolite screening between the above-ground parts and the underground parts.

#### 2.3.3. Orthogonal Projections to Latent Structure-Discriminant Analysis (OPLS-DA)

OPLS-DA is a multivariate statistical method with supervised pattern recognition, which can effectively eliminate irrelevant effects and screen differential metabolites. According to the results of principal component analysis and cluster analysis (Figure 6, the variability of the roots, stems, and leaves of *Houttuynia cordata* was mainly in the first principal component. To prevent the model from overfitting, the permutation test (100 response ranking test) was used to evaluate the model. The R2Y and Q2 values of the model were greater than 0.9, indicating that the model had high predictive power and the model was meaningful. Therefore, differential metabolites can be screened according to variable importance projection (VIP).

### 2.4. Screening and Identification of Different Metabolites

#### 2.4.1. Screening for Differential Metabolites

The T-test was combined with VIP values of OPLS-DA (*p* < 0.05 and VIP > 1) to screen the differential metabolites of different parts. Meanwhile, volcanic maps of different metabolites were drawn according to VIP and fold variation values (Figure 7). The statistics of significant differences in metabolites of different parts of Houttuynia cordata are shown in Table 1. There were more differential metabolites in the roots compared to the stems and leaves and more up-regulated metabolites in the roots. There were not many differential metabolites between stems and leaves, and they were more significantly up-regulated in leaves.

#### 2.4.2. Distribution of Metabolite Components in Different Parts

According to the Wayne diagram (Figure 8A), there were 45 shared differential metabolites in the stems and leaves, and roots, the specific differential metabolites are shown in Table 2. The cluster analysis resulted in 28 metabolites with up-regulation and 17 metabolites with down-regulation in the roots. The stems and leaves were clustered into one cluster and the roots into one cluster (Figure 8B).

#### 2.4.3. Differential Metabolic Chemical Structure and Function Analysis of Aboveground Parts and Underground Parts

From multivariate statistical analysis and cluster analysis, the above-ground parts (stems and leaves) differ greatly from the underground parts (roots). Therefore, this study focused on analyzing the differential metabolites between above-ground and underground parts. In Figure 9, the screening looked for 14 metabolites with pharmacological effects, six of which were up-regulated in the underground part, including 10-nonadecanone, 2-undecanone, 3-carene, 2-carene, 2-tridecanone and Citral. In the above-ground part, eight metabolites were up-regulated, including Nonanal, 1-decanol, 1-dodecanol, Tridecanal, 2-undecenal, Pentanal, 1-penten-3-ol and Orcinol.

## 3. Discussion

### 3.1. Different Parts Food Value and Differential Metabolites

*Houttuynia cordata* is both edible and medicinal [25]. In China, people only consume its roots and discard the stems and leaves [10]. This is very wasteful, and *Houttuynia cordata* is becoming scarce in the wild and less cultivated in captivity. Therefore, we need to explore how to use the resources of *Houttuynia cordata* wisely to maximize its value. The medicinal parts are generally the whole herb or fresh above-ground parts [26], but the difference between the two is not studied by scholars at present. We explored the potential medicinal value and utilization areas of different parts by studying the differences in metabolites of different parts. The nutritional value of *Houttuynia cordata* as a wild vegetable is high, and this study showed that the roots of *Houttuynia cordata* are rich in vitamin C, and the leaves are high in total flavonoids, total protein, and soluble sugars. However, the edibility value of the stems is not high, and there is significant variability in the nutritional composition of the roots and leaves. Rational use of different parts is especially important, and although the edibility value of stems is not high, their medicinal properties need to be utilized. Many scholars use medicinal plants to be used as animal feed additives [27]. For example, the dried and crushed above-ground parts of *Houttuynia cordata* can replace antibiotics, the antibacterial ability of the herb can enhance the growth performance of chickens and ducks [28], and the powder of *Houttuynia cordata* can be effectively used in agriculture, giving the above-ground parts a high economic value.

The clustering analysis of metabolites revealed that the stems and leaves were clustered into one large cluster and the roots were clustered into one large cluster so that the metabolic variability between the above-ground and below-ground parts of *Houttuynia cordata* could be studied. A total of 45 differential metabolites were found, of which 17 were up-regulated in the above-ground part and 28 in the roots. Among the underground parts, the metabolites with relatively high contents were 10-nonadecanone, 2-undecanone, 3-carene, 2-carene, 2-tridecanone and Citral. In the underground part, eight metabolites were up-regulated, including Nonanal, 1-decanol, 1-dodecanol, Tridecanal, 2-undecenal, Pentanal, and 1-penten-3-ol.

### 3.2. Pharmacological Effects of Differential Metabolites in Different Parts

The focus of this study is to investigate the distribution and pharmacological effects of the metabolites in different parts of *Houttuynia cordata*. Among them, the active components in the roots were mainly olefins, esters, and alcohols, in the stems they were mainly esters, alkanes, and aldehydes, and in the leaves were mainly alkanes, aldehydes, lipids, and alcohols. In order to express the distribution of these metabolites in different parts, cluster analysis was used, and it was found that the metabolites of *Houttuynia cordata* were divided into two categories: above-ground and underground. Meanwhile, the differential metabolites of different parts were screened by using the t-test combined with the VIP value of OPLS-DA (*p* < 0.05 and VIP > 1). The clustering of metabolites common to different parts showed that 28 metabolites were up-regulated and 17 metabolites were down-regulated in the underground part. To explore the potential value of above-ground versus below-ground parts, we analyzed the pharmacological effects of metabolites from different parts.

Headspace solid-phase micro-extraction (HS-SPME) is a new technology; its main advantage is that it can eliminate interference from the sample matrix [23]. *Houttuynia cordata* has many pharmacologically active components, so the medicinal value of the differential metabolites of the above-and below-ground parts of *Houttuynia cordata* was analyzed [29]. A significantly up-regulated metabolite 2-undecanone was found in the below-ground part, which indicates that it is mainly clustered in the roots. 2-undecanone is the most important compound in *Houttuynia cordata*. It has good antioxidant, anti-inflammatory [30], antiviral, immunomodulatory, antibacterial, and anti-tumor effects, and can also be used as a mosquito repellent [31,32]. Some scholarly studies have shown that 2-Undecanone can significantly inhibit RAW264.7 macrophage production of inflammatory cytokines including tumor necrosis factor-α (TNF-α) and interleukin-1β (IL-1β) upon lipopolysaccharide (LPS)-mediated activation of the Toll-like receptor 4 (TLR4) signaling pathway [33]. 2-undecanone has a wide range of pharmacological effects with anti-inflammatory properties; for example, it was found that 2-undecanone prevents inflammation caused by PM2.5, and other mechanisms need to be further studied [34]. Therefore, the root can be mainly used as the extraction site when extracting 2-undecanone. In addition, other differential metabolites of the root also have some medicinal effects. 3-carene has anti-inflammatory, antibacterial, anti-anxiety, and sleep-enhancing effects [35]; 2-carene has anti-inflammatory and antibacterial effects [36]; 2-tridecanone can be used as an anthelmintic [37]; Citral has anti-inflammatory and [38] bactericidal [39] effects; and 10-nonadecanone can be used as an organic solvent [40].

Among the differential metabolites in the underground part that have edible and medicinal value are Nonanal, 1-decanol, 1-dodecanol, Tridecanal, 2-undecenal, Pentanal, 1-penten-3-ol, and Orcinol. 2-undecenal is an aroma compound that can be used as an edible flavor [41] and has some antifungal and [42] antioxidant abilities [43]. Tridecanal can be used as a food additive capable of providing flavor to food [44] and has some antibacterial effects [45]. 1-dodecanol can act as a surfactant [46] and is immune to the toxicity of Aedes aegypti [47]; it has some antioxidants [48] and insecticidal effects [47]. 1-decanol can be used as a chemical pruning agent to inhibit the growth of axillary buds [49] and has some fungal inhibitory effects [50]. Nonanal has anti-inflammatory, antibacterial [51], and antidiarrheal [52] effects and is also used in fragrances [53], Pentanal and 1-penten-3-ol are mainly used as fragrances [54,55], and Orcinol has antioxidant properties [56].

In summary, most of the main components of the underground part have pharmacological effects such as anti-inflammatory, anti-bacterial and antiviral, which are more suitable for drug development. Furthermore, the above-ground part has more spice components and good antioxidant capacity, which is suitable for the extraction of edible flavors. The mechanism of medicinal use of *Houttuynia cordata* is still unclear, and the main products are volatile oils, etc. [57]. Since the extraction of *Houttuynia cordata* mainly followed the mixing of roots and stems and leaves, our experimental results show that the medicinal components of different parts varied greatly. The mixed extraction may affect its medicinal activity and therapeutic effect. For example, in this study, 2-undecanone, a medicinal component, was mainly distributed in the underground part. Whether the mixed extraction will affect its therapeutic effect compared to root extraction requires more in-depth study by subsequent scholars.

## 4. Materials and Methods

### 4.1. Experimental Design

The experiment was conducted in the greenhouse of Shanghai Sunqiao Yijia Co., Ltd. at an average temperature of 20 °C. The experiment lasted from 18 March 2021 to 20 July 2021. *Houttuynia cordata* seedlings were divided on 18 March 2021, selected for robust growth and consistent growth; side shoots were removed and planted in a cultivation pond with drip irrigation and sand culture, and 3–4 plants were planted at 10 cm intervals in the pond. At the maturity of the plants, a certain amount of fishy grass was randomly harvested and the total protein, soluble sugar, vitamin C contents, and total flavonoid contents of different parts of the fishy grass were measured to determine the edibility value of different parts, and each test was repeated three times to take the average value.

Non-targeted GC-MS was performed by headspace solid-phase microextraction (HS-SPME-GC-MS) on different parts of *Houttuynia cordata*; the metabolites of the different parts were identified and analyzed using multivariate statistical methods to determine the medicinal value of the different parts, as shown in Figure 10. The experiment mainly investigated the differences between above-ground and below-ground parts for edible and medicinal purposes and rationalized the use of different parts of *Houttuynia cordata* to maximize its medicinal value.

### 4.2. Experimental Material

The *Houttuynia cordata* variety was Sunqiao No.1, planted in a planting tank with 100% sand substrate, and was watered with Hoagland nutrient solution, which consisted of N(NO_3_^−^) 15 mmol/L, P 1 mmol/L, K 6 mmol/L, Ca 5 mmol/L, Mg 2 mmol/L, and S 2 mmol/L as the macronutrients. Both the test variety and the test nutrient solution were provided by Shanghai Yijia Agricultural Technology Co.

*Houttuynia cordata* seedlings were cultivated using cavity trays. On 18 March 2021, healthy and uniformly growing *Houttuynia cordata* seedlings were selected for division, and side shoots were removed and transplanted into a planting tank 5 m long, 0.4 m wide, and 0.3 m deep. The average temperature of the greenhouse was 20 °C. Fertilization of *Houttuynia cordata* was carried out by drip irrigation, with the pH of the nutrient solution controlled at 6 and the EC value controlled at about 0.3. Water and fertilizer were drip irrigated every 5 min 500 mL at a time.

### 4.3. Method for Determination of Nutritional Indexes

Total flavonoid content was determined by the ultrasonic extraction method (kit provided by Congye Bio) according to the manufacturer’s instructions.Vitamin C was determined by the phosphomolybdic acid colorimetric method (kit provided by Shanghai Yuan Ye Biotechnology Co.) according to the manufacturer’s instructions.Total protein content was determined by the Komas Brilliant Blue method (kit provided by Nanjing Jiancheng Institute of Biological Engineering) according to the manufacturer’s instructions.Soluble sugar was determined by the anthrone colorimetric method (kit provided by Nanjing Jiancheng Institute of Biological Engineering) according to the manufacturer’s instructions.

### 4.4. Determination Method of Volatile Components

#### 4.4.1. Sample Pretreatment

About 4 g of the sample was weighed and transferred to a 20 mL headspace injection vial for HS-SPME-GC-MS analysis.

#### 4.4.2. Analysis Conditions of GC-MS

##### HS-SPME Extraction Conditions

The sample was shaken for 15 min at a constant temperature of 60 °C at 450 rpm (5 s on, 2 s off), and the 50/30 µM DVB/CAR on the PDMS extraction head was inserted into the headspace portion of the sample and extracted in the headspace for 60 min, resolved at 250 °C for 5 min, and then separated and identified by GCMS with the extraction head. The extraction head was aged in a Fiber Conditioning Station for 2 h before extraction, and the sample was desorbed by heating in the Fiber Conditioning Station for 10 min before and after sampling.

##### Chromatographic Condition

We used a DB-WAX capillary column (30 m × 0.25 mm × 0.25 μM, Agilent J&W Scientific, Folsom, CA, USA) with high purity helium (purity not less than 99.999%) as the carrier gas, a constant flow rate of 1.0 mL/min, an inlet temperature of 260 °C, no split injection, and a solvent delay of 1.5 min. Procedure temperature rise: 40 °C for 3 min, 5 °C/min to 220 °C for 5 min.

##### Mass Spectrometry Conditions

We used an electron bombardment ion source (EI) with an ion source temperature of 230 °C, quadrupole temperature of 150 °C, and electron energy of 70 eV. The scan mode was full scan mode (SCAN), mass scan range: *m/z* 20–650.

#### 4.4.3. Qualitative Analysis

MS-DIAL performed a series of processes on the imported data such as peak detection, peak identification, MS2Dec deconvolution, characterization, peak alignment, filtering, missing value interpolation, etc. Metabolite characterization was based on the LUG database. The final raw data matrix was derived. The National Institute of Standards and Technology (NIST) database (https://webbook.nist.gov/chemistry/ (accessed on 15 August 2021) was also used for substance characterization.

### 4.5. Data Processing and Analysis

To analyze the difference in edible indicators, a one-way ANOVA was performed using SPSS 22.0 software, Origin Pro 2021 was used for graphing and AI was used for layout. The metabolic data were analyzed using a non-targeted metabolomics processing flow, and the raw data matrix was normalized. Simca 14.1 was imported, orthogonal partial least squares discriminant analysis software (OPLS-DA) was used, and the conditions of VIP > 1 and *p* < 0.05 were selected to screen for differential metabolites. Cluster analysis, Wayne analysis, and principal component analysis (PCA) were performed using R 3.6.1.

## 5. Conclusions

To explore the edible and medicinal value of different parts of *Houttuynia cordata*, and to maximize the use of resources. It was found that the roots and leaves had higher edibility values and were significantly different from the stems. The metabolites of different parts were determined by HS-SPME/GC-MS, and OPLS-DA and cluster analysis were used to find the differential metabolites of different parts, and it was found that the different parts of *Houttuynia cordata* were divided into two major clusters—above-ground and underground. Among the differential metabolites, there were eight species with pharmacological effects in the above-ground part and six species in the underground part. It was found that the important medicinal component 2-undecanone was mainly distributed in the underground part, so the underground part could be targeted for efficient extraction. Most of the main components of the underground part have pharmacological effects such as anti-inflammatory, anti-bacterial and antiviral, which are more suitable for drug development. Furthermore, the above-ground part has more spice components and good antioxidant capacity, which is suitable for the extraction of edible flavors. The rational utilization of different parts can bring out the maximum value of *Houttuynia cordata*.

## Figures and Tables

**Figure 1 molecules-27-03921-f001:**
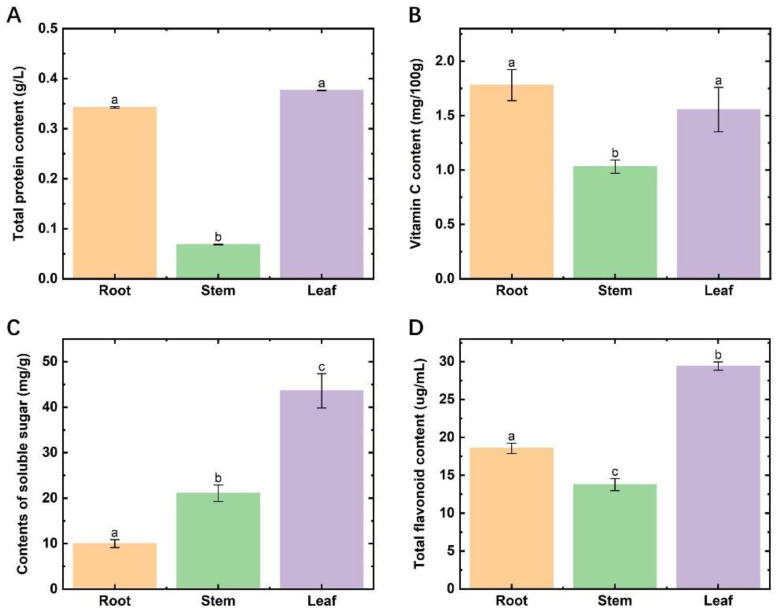
Nutrient contents in different parts of Houttuynia cordata. (**A**) the content of total protein; (**B**) the content of vitamin C; (**C**) the content of soluble sugar; (**D**) the content of total flavonoids. Different lowercase letters indicate a significant difference (*p* < 0.05).

**Figure 2 molecules-27-03921-f002:**
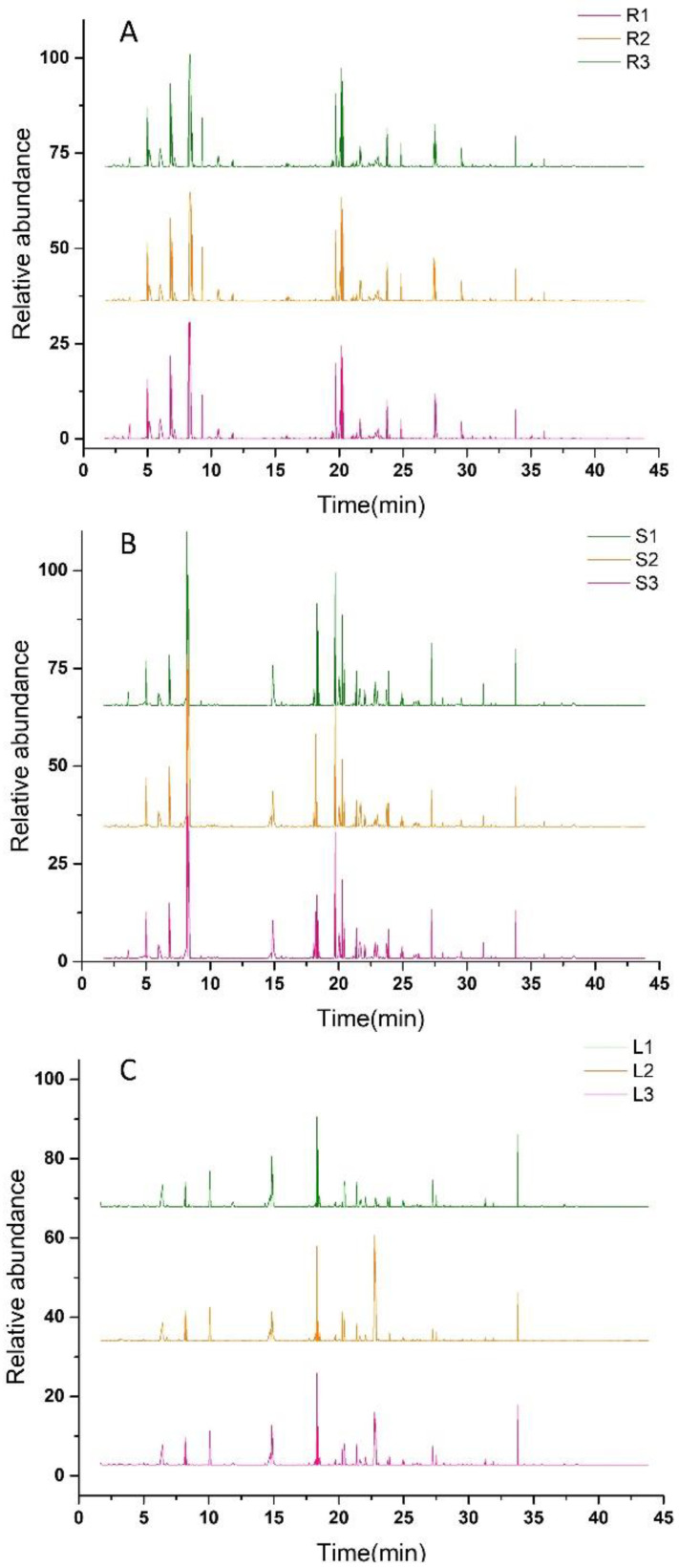
GC–MS chromatogram of *Houttuynia cordata*. (**A**) represents the total ion flow diagram of the *Houttuynia cordata* root; (**B**) represents the total ion flow diagram of the *Houttuynia cordata* stem; (**C**) represents the total ion flow diagram of the *Houttuynia cordata* leaf.

**Figure 3 molecules-27-03921-f003:**
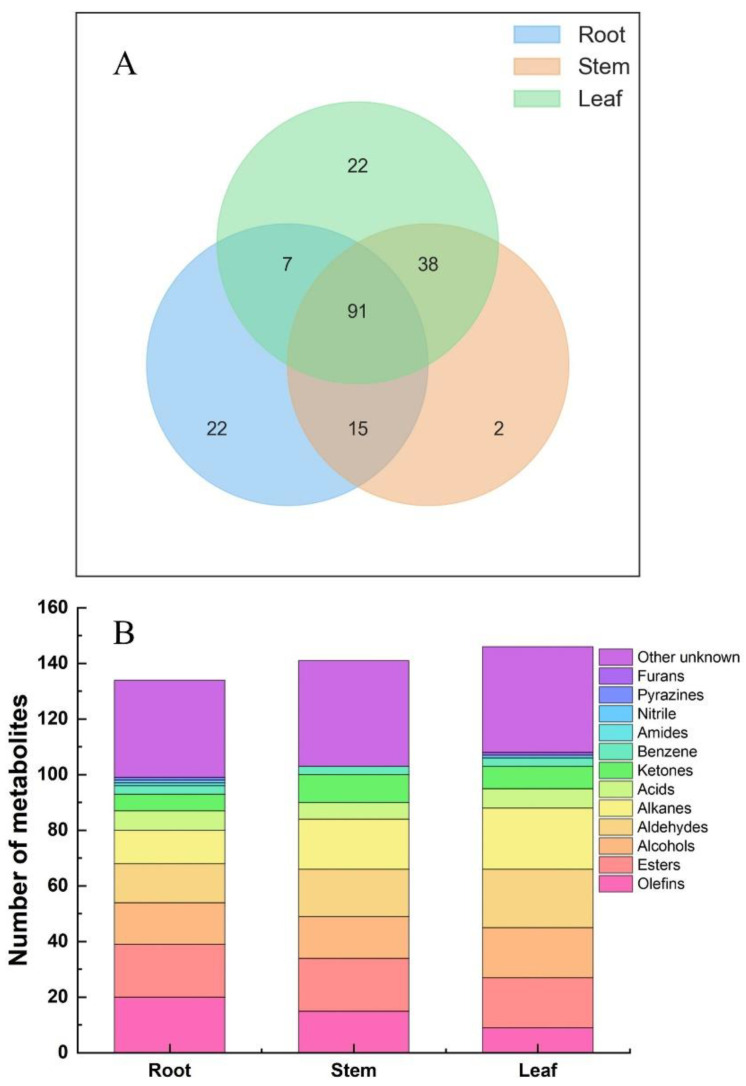
Metabolites of different parts of *Houttuynia cordata*. (**A**) Wayne diagram of metabolites from Different parts of *Houttuynia cordata*; (**B**) Columnar accumulation diagram of different metabolites in *Houttuynia cordata*.

**Figure 4 molecules-27-03921-f004:**
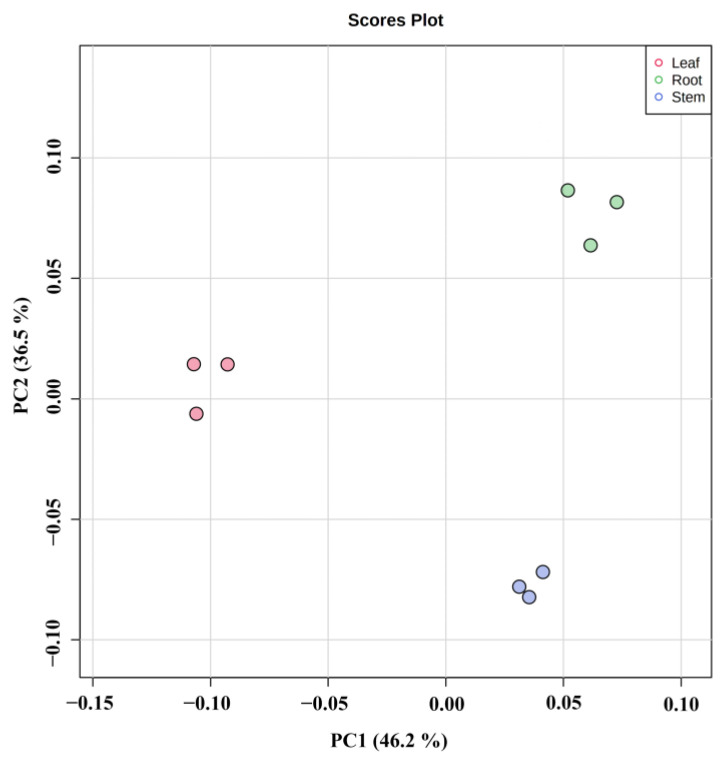
PCA score plot in different parts of *Houttuynia cordata*.

**Figure 5 molecules-27-03921-f005:**
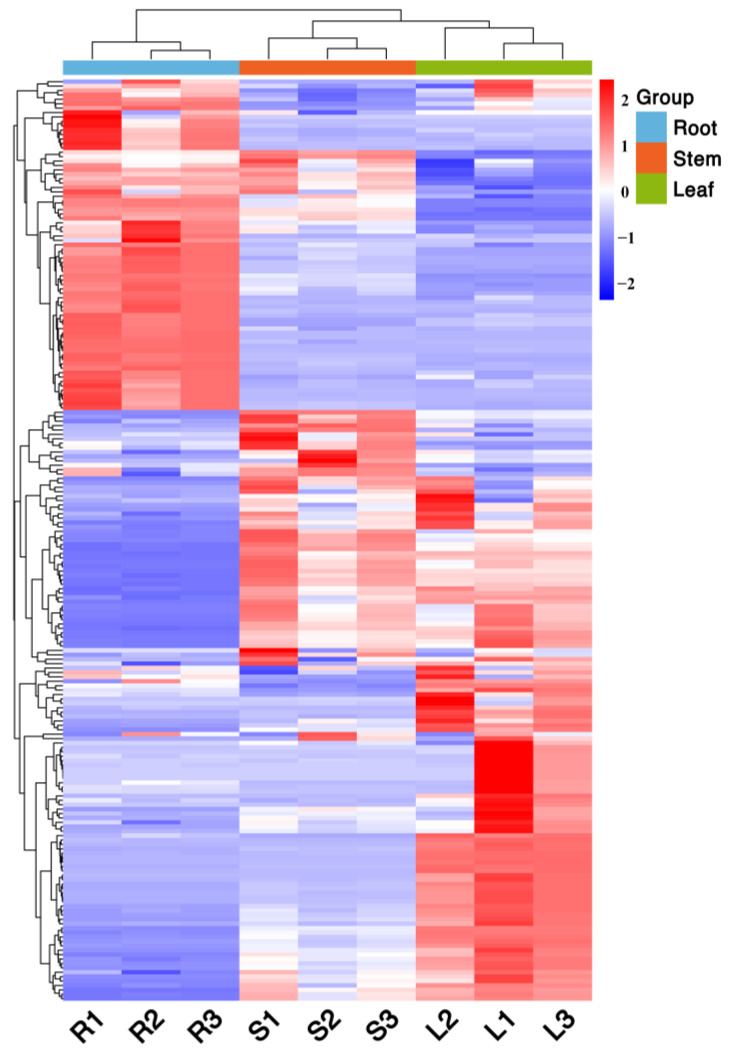
Cluster map of metabolites in different parts of *Houttuynia cordata*. The color sequence from red to blue indicates metabolites abundance from high to low.

**Figure 6 molecules-27-03921-f006:**
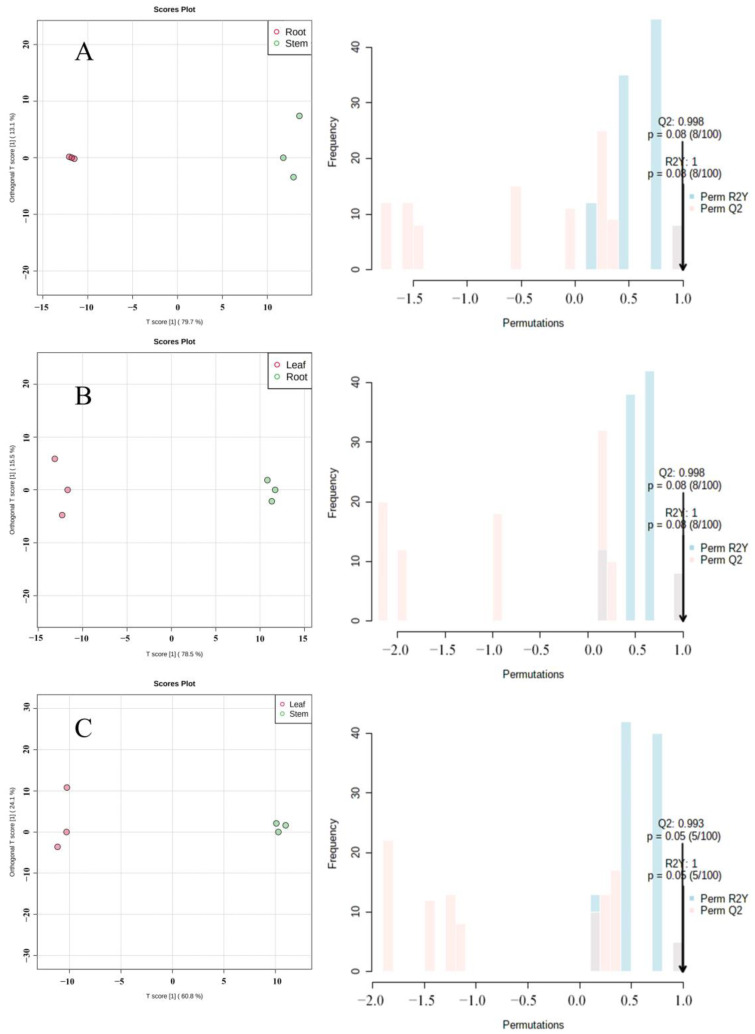
Orthogonal partial least squares regression analysis (OPLS-DA) score chart and 100 response ranking tests. (**A**) Root vs. stem; (**B**) root vs. leaf; (**C**) leaf vs. stem.

**Figure 7 molecules-27-03921-f007:**
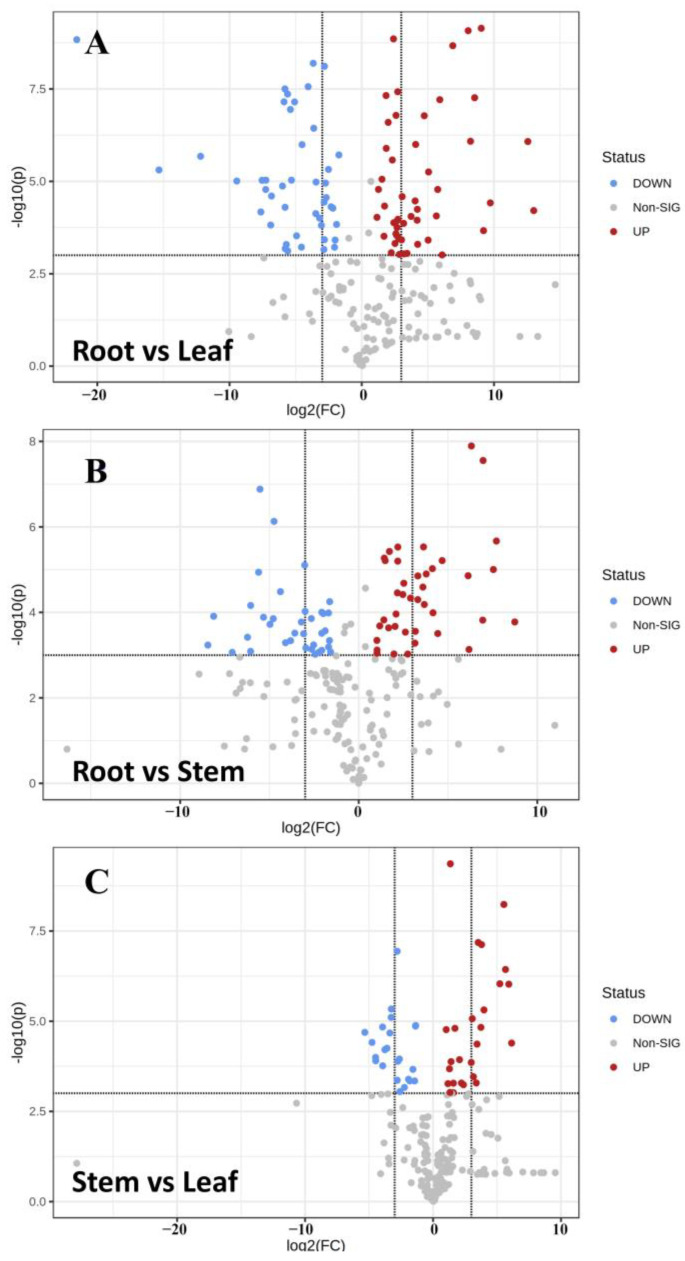
Volcanic map of differential metabolites in different parts of *Houttuynia cordata*. (**A**) Root vs. leaf; (**B**) root vs. stem; (**C**) stem vs. leaf. The abscissa represents the sample name, and the ordinate represents the differential metabolite. The color from blue to red indicates the expression abundance of metabolites from low to high, that is, the redder the color, the higher the expression abundance of differential metabolites.

**Figure 8 molecules-27-03921-f008:**
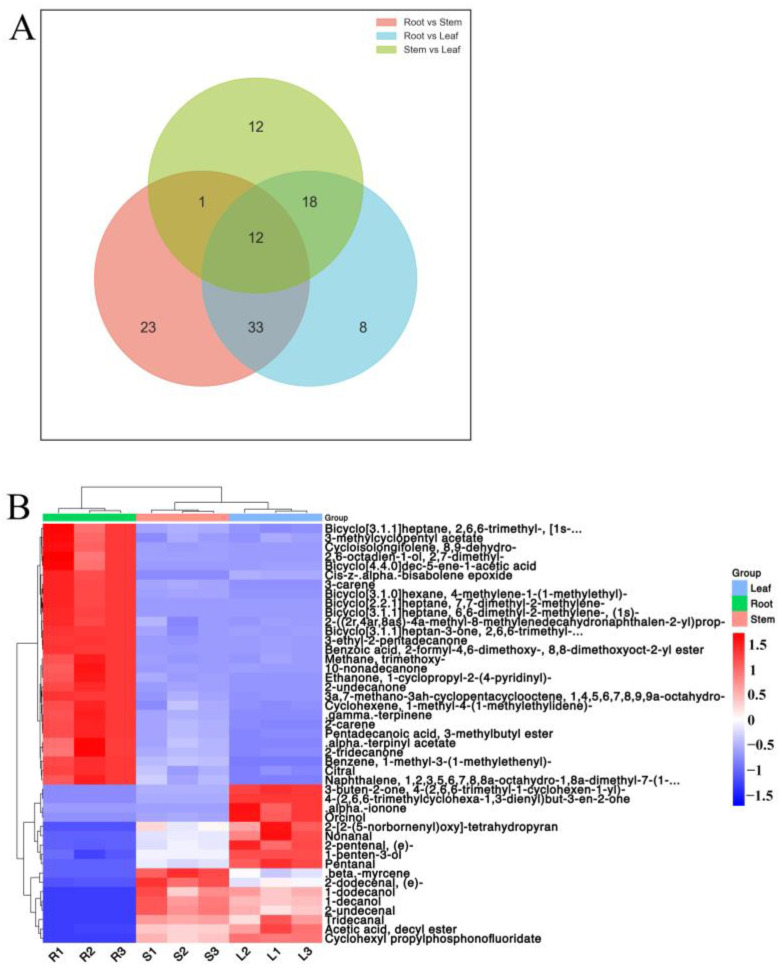
Specific differential metabolites in different parts of *Houttuynia cordata*. (**A**) heat map of differential metabolites. The color sequence from red to blue indicates metabolites abundance from high to low; (**B**) Wayne diagram.

**Figure 9 molecules-27-03921-f009:**
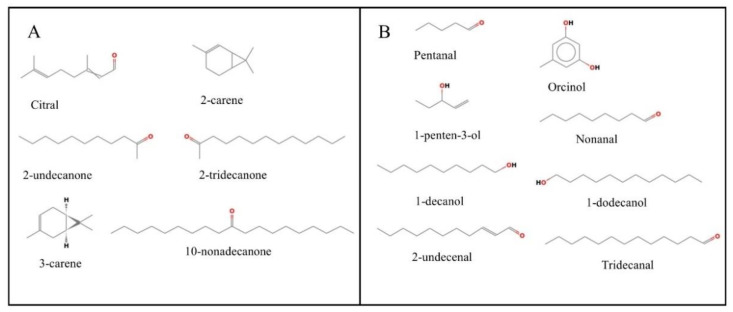
Chemical structures of differential metabolites of *Houttuynia cordata*. (**A**) Six metabolites from underground parts (roots); (**B**) Eight metabolites from above-ground parts (stems and leaves).

**Figure 10 molecules-27-03921-f010:**
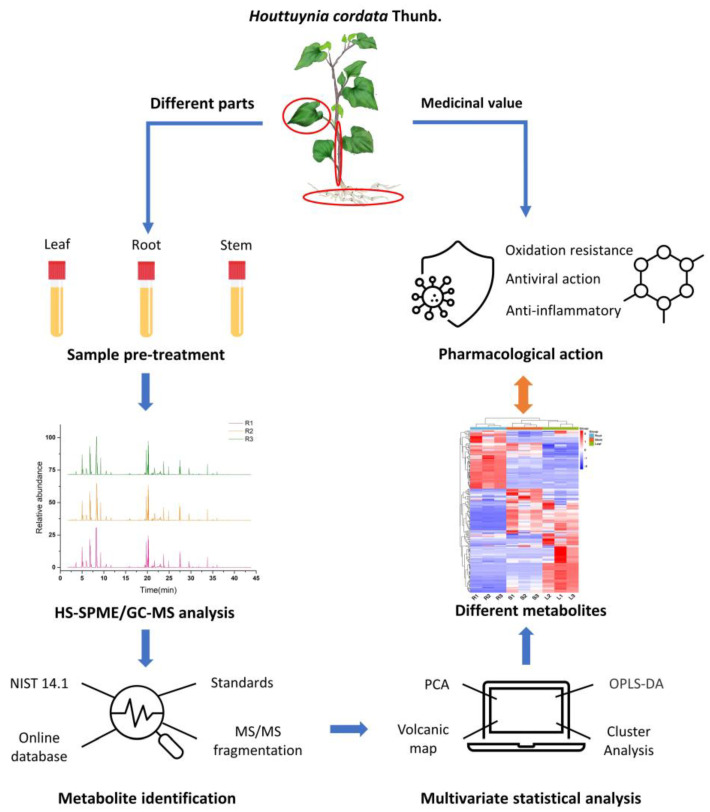
Flow chart of *Houttuynia cordata* test.

**Table 1 molecules-27-03921-t001:** Statistical table of the number of metabolites in different parts of *Houttuynia cordata*.

Group Name	All Sig Diff	Down-Regulated	Up-Regulated
Root vs. Stem	69	38	31
Root vs. Leaf	71	42	29
Stem vs. Leaf	44	18	26

**Table 2 molecules-27-03921-t002:** Identification results of differential metabolites from different parts of *Houttuynia cordata*.

No.	Time (min)	Metabolite Name	Quant Mass	Ionic Strength		
				Root	Stem	Leaf
1	4.253	Pentanal	44.02	0.12	0.71	1.64
2	5.166	3-carene	93.05	76.35	5.88	0.38
3	5.689	Methane, trimethoxy-	75.02	1.33	0.08	0.08
4	6.905	Bicyclo[3.1.1]heptane, 6,6-dimethyl-2-methylene-, (1s)-	69.10	288.24	1.06	0.05
5	7.126	Bicyclo[3.1.0]hexane, 4-methylene-1-(1-methylethyl)-	77.02	40.74	2.27	0.16
6	7.662	2-[2-(5-norbornenyl)oxy]-tetrahydropyran	85.04	0.01	0.98	1.82
7	7.672	2-pentenal, (e)-	83.02	0.01	0.30	0.75
8	8.186	beta-myrcene	93.06	0.04	603.16	229.14
9	8.422	Bicyclo[2.2.1]heptane, 7,7-dimethyl-2-methylene-	91.06	145.11	0.60	0.15
10	8.614	3-methylcyclopentyl acetate	72.04	0.56	0.05	0.05
11	9.08	1-penten-3-ol	57.03	0.14	0.90	1.86
12	9.274	Ethanone, 1-cyclopropyl-2-(4-pyridinyl)-	93.04	227.95	9.18	0.93
13	10.532	gamma-terpinene	93.05	53.25	5.49	0.34
14	11.667	Cyclohexene, 1-methyl-4-(1-methylethylidene)-	121.07	33.93	3.11	0.18
15	11.977	2-carene	121.07	1.88	0.26	0.02
16	14.853	Nonanal	57.06	3.52	129.90	320.80
17	15.889	Benzene, 1-methyl-3-(1-methylethenyl)-	132.06	18.40	4.25	0.59
18	16.583	Bicyclo[3.1.1]heptane, 2,6,6-trimethyl-, [1s-(1.alpha,2.beta,5alpha)]-	55.02	1.81	0.21	0.06
19	17.489	3a,7-methano-3ah-cyclopentacyclooctene, 1,4,5,6,7,8,9,9a-octahydro-1,1,7-trimethyl-, [3ar-(3aalpha,7alpha.,9abeta)]-	161.10	0.53	0.03	0.01
20	18.03	Benzoic acid, 2-formyl-4,6-dimethoxy-, 8,8-dimethoxyoct-2-yl ester	192.96	3.87	0.02	0.06
21	18.71	Bicyclo[3.1.1]heptan-3-one, 2,6,6-trimethyl-, (1.alpha,2beta,5alpha)-	83.06	1.57	0.06	0.09
22	19.274	Orcinol	124.02	0.03	0.71	9.52
23	20.171	2-undecanone	58.06	459.57	14.02	0.01
24	20.244	10-nonadecanone	71.06	315.64	2.09	2.10
25	22.021	Acetic acid, decyl ester	43.03	1.60	48.58	67.01
26	22.325	alpha-terpinyl acetate	121.07	18.47	2.79	0.26
27	22.909	Naphthalene, 1,2,3,5,6,7,8,8a-octahydro-1,8a-dimethyl-7-(1-methylethenyl)-, [1r-(1alpha,7beta,8aalpha)]-	119.06	12.29	2.28	0.17
28	23.141	Citral	82.04	1.91	0.34	0.12
29	23.603	2-undecenal	70.05	0.01	1.71	1.27
30	23.892	1-decanol	70.06	1.98	98.73	77.85
31	24.845	2-tridecanone	58.04	109.27	15.60	1.40
32	24.972	Tridecanal	57.04	0.20	42.09	42.94
33	25.695	alpha-ionone	121.04	0.06	0.35	12.69
34	25.879	2-dodecenal, (e)-	70.03	0.87	11.21	5.76
35	27.509	Cis-z-alpha-bisabolene epoxide	43.06	198.36	2.14	22.52
36	27.527	3-buten-2-one, 4-(2,6,6-trimethyl-1-cyclohexen-1-yl)-	177.10	0.20	9.81	84.23
37	28.129	1-dodecanol	55.04	0.41	20.48	18.07
38	28.766	4-(2,6,6-trimethylcyclohexa-1,3-dienyl) but-3-en-2-one	175.07	0.02	0.10	0.76
39	29.667	2,6-octadien-1-ol, 2,7-dimethyl-	69.05	16.52	0.26	0.07
40	30.442	Bicyclo [4.4.0] dec-5-ene-1-acetic acid	134.05	11.95	0.12	0.17
41	31.778	3-ethyl-2-pentadecanone	86.01	13.54	0.12	0.22
42	35.029	Pentadecanoic acid, 3-methylbutyl ester	70.03	13.70	1.83	0.24
43	36.15	Cycloisolongifolene, 8,9-dehydro-	202.15	2.33	0.08	0.05
44	39.889	2-((2r,4ar,8as)-4a-methyl-8-methylenedecahydronaphthalen-2-yl) prop-2-en-1-ol	95.09	0.66	0.08	0.07
45	41.506	Cyclohexyl propylphosphonofluoridate	126.99	0.09	0.99	1.27

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
