# Peer review of "Quality and Metabolomics Analysis of Houttuynia cordata Based on HS-SPME/GC-MS"

_molecules, 2022, doi:10.3390/molecules27123921_

Round 1

Reviewer 1 Report

The paper by Shuai Qi, Lingyan Zha, Yongzheng Peng, Wei Luo, Kelin Chen, Xin Li, Danfeng Huang and Dongmei Yin entitled “Quality and Metabolomics Analysis of Houttuynia cordata Based on HS-SPME/GC-MS”.

The results were presented in a way that is not very easy to understand. The authors could improve the writing of this part.

Furthermore, the discussion of metabolomics analysis should be deepened, addressing better the compounds responsible for the discrimination of the samples. They are shown in Figure 8, but reading the names is practically impossible.

Is the use of dots in compound names correct? Example: .alpha.-ionone.

In general, the subject covered is interesting, and a good presentation of the article will interest many people.

Author Response

Thank you for your suggestions, all of them are very important and they will guide me in my future research work.

Point 1:

Referee: The results were presented in a way that is not very easy to understand. The authors could improve the writing of this part.

Reply: The structure of the results section has been adjusted to make the reader's reading feel more fluid.

Point 2:

Referee: Furthermore, the discussion of metabolomics analysis should be deepened, addressing better the compounds responsible for the discrimination of the samples. They are shown in Figure 8, but reading the names is practically impossible.

Reply: Thanks to your comments, we have deepened our discussion on the metabolomics section.

Point 3:

Referee: Is the use of dots in compound names correct? Example: .alpha.-ionone.

Reply: The dots in the compound names were derived from the NIST database, And the use was not very standardized. We made full text changes. Your comments are greatly appreciated.

Thanks again for your advice and I hope to learn more from you.

Reviewer 2 Report

The manuscript, entitled "Quality and Metabolomics Analysis of Houttuynia cordata Based on HS-SPME/GC-MS", is an interesting and extensive work on potentially biologically active compounds found in Houttuynia cordata. The determinations were made using three parts of the plant: roots, stem, and leaves. The authors present interesting comparative analyzes of the compounds present in the above-ground and underground parts of this species. The work is well planned, and every part of it is described in detail. Modern analytical methods, as well as comparative and statistical analysis of the results, draw attention.

Some figures are poorly visible and require correction of the font or general enlargement before publication, like 3/B, 4, 6, 7, and 8.

Author Response

Thank you for your suggestions, all of them are very important and they will guide me in my future research work.

Point 1

Referee: Some figures are poorly visible and require correction of the font or general enlargement before publication, like 3/B, 4, 6, 7, and 8.

Reply: Thanks to your suggestions, we have adjusted the figure layout and font size to make the figures more readable.

Thanks again for your advice and I hope to learn more from you.
